

# Evaluation of genetic variation and host resistance to wheat stem rust pathogen (*Puccinia graminis* f. sp. *tritici*) in bread wheat (*Triticum aestivum* L.) varieties grown in Türkiye

Ahmet Cat

Department of Plant Protection, Siirt University, Siirt, Türkiye

## ABSTRACT

Wheat stem rust, which is caused by *Puccinia graminis* f. sp. *tritici* (*Pgt*), is a highly destructive disease that affects wheat crops on a global scale. In this study, the reactions of 150 bread wheat varieties were evaluated for natural *Pgt* infection at the adult-plant stage in the 2019–2020 and 2020–2021 growing seasons, and they were analyzed using specific molecular markers to detect stem rust resistance genes (*Sr22*, *Sr24*, *Sr25*, *Sr26*, *Sr31*, *Sr38*, *Sr50*, and *Sr57*). Based on phenotypic data, the majority of the varieties (62%) were resistant or moderately resistant to natural *Pgt* infection. According to molecular results, it was identified that *Sr57* was present in 103 varieties, *Sr50* in nine varieties, *Sr25* in six varieties, and *Sr22*, *Sr31*, and *Sr38* in one variety each. Additionally, their combinations *Sr25* + *Sr50*, *Sr31* + *Sr57*, *Sr38* + *Sr50*, and *Sr38* + *Sr57* were detected in these varieties. On the other hand, *Sr24* and *Sr26* were not identified. In addition, many varieties had low stem rust scores, including a large minority that lacked *Sr57*. These varieties must have useful resistance to stem rust and could be the basis for selecting greater, possibly durable resistance.

## INTRODUCTION

Wheat (*Triticum* L.) is one of the most important cereal crops grown worldwide due to its contribution to human nutrition as a 20% main energy source; it also contains substantial vitamins and minerals for human health. Owing to the increasing world population every year and limited production areas, food shortages are unavoidable. Fungal diseases have accelerated this trend. Stem rust caused by the biotrophic fungal pathogen *Puccinia graminis* f. sp. *tritici* (*Pgt*) is one of the most threatening wheat diseases, causing huge yield losses worldwide. It was reported that yield losses of up to 80% have occurred during epidemic conditions (*Figueroa, Hammond-Kosack & Solomon, 2018*). In the past, growing resistant varieties and the United States and Europe are the main countries that use the eradication of *Berberis* spp. as an effective method to control wheat stem rust, rather than most other countries. *Peterson et al. (2005)* except for Ethiopia where the pathogen caused dramatic yield losses of the wheat variety "Enkoy" in 1993 and 1994 (*Shank, 1994*;

Corresponding author
Ahmet Cat, ahmetcat@siirt.edu.tr

*Singh et al., 2011*). Over the past two decades, there have been reports of aggressive pathogenic strains classified as TTKSK. For instance, in 1998, a new emergent race known as Ug99, designated TTKSK in Uganda, was discovered to have broken down the resistance gene *Sr31* (*Pretorius et al., 2000*). Since then, more than 17 genes have been rendered ineffective against Ug99 (*Singh et al., 2008*). Subsequently, it was reported that new variants of Ug99 have evolved and exhibit virulence against the resistance genes *Sr24*, *Sr36*, *Sr38*, and *SrTmp* (*Jin et al., 2008*, *2009*; *Newcomb et al., 2016*). Additionally, Ug99 race and its variants have been determined in different countries especially in Africa where they caused severe outbreaks of the pathogen (*Fetch et al., 2016*; *Nazari et al., 2021*; *Singh et al., 2011*). In 2016, the stem rust race TTTTF caused the breakdown of resistance in numerous durum and bread wheat varieties in Sicily, representing a significant threat that has been identified in other European countries (*Bhattacharya, 2017*; *Olivera Firpo et al., 2017*; *Olivera et al., 2015*, *2019*). Wheat varieties carrying resistance genes are used to effectively control of stem rust disease. The use of resistant varieties of stem rust pathogens is essential; however, its variants can overcome resistance genes. To prevent this from happening, it is necessary to obtain knowledge about the gene content of wheat varieties and their respective frequencies Generally, rust resistance genes in wheat have been grouped into two groups, conferring forms all-stage or seedling resistance (ASR) and adult plant resistance (APR).

Molecular markers have been developed and routinely used to study the genetic resistance of varieties, varieties or landraces against stem rust disease (*Goutam et al., 2015*). So far, 70 stem rust resistance genes have been reported in wheat and their relatives (*McIntosh et al., 2016*). In this study, *Sr* resistance genes, specifically *Sr22*, *Sr24*, *Sr25*, *Sr26*, *Sr31*, *Sr38*, and *Sr57* provide resistance to stem rust at all stages, while with Sr57 demonstrating effectiveness during the adult plant stage (APR). However, *Pgt* pathotypes that are virulent on some resistance genes have been reported. For example; *Jin et al. (2008)* reported the occurrence of susceptible infection responses in wheat lines and cultivars carrying *Sr24* during a stem rust screening field nursery conducted in Njoro, Kenya. In addition, one isolate showing virulence on *Sr50* was detected in 2014 (*Mago et al., 2015*). Nevertheless, all three resistance genes, namely *Sr22*, *Sr26*, and *Sr50*, remain effective against TTKSK (Ug99) and its derivatives (*Singh et al., 2011*). *Abou-Zeid et al. (2023)* stated that the resistance genes *Sr22*, *Sr25*, *Sr26*, and *Sr50* may currently be the most helpful race-specific resistance genes, depending on how they are used in combination. Most previous studies on this subject have been carried out by various researchers who have identified the frequency of different *Sr* resistance genes in wheat varieties from different countries (*Baranova et al., 2023*; *Kelbin et al., 2022*; *Li et al., 2016*; *Lin et al., 2021*; *Xu et al., 2018*, *2017*).

Türkiye is one of the major wheat-producing countries, with 17.650 million tons of production and nearly 6.7 million ha acreage (*FAO, 2020*). It is also one of the most important gene centers in wheat. Many stem rust epidemics occurred in Türkiye from 1932 to 1960. High yield losses, ranging from 40% to 75%, were recorded in the Mediterranean and Aegean regions (*İyriboz & İleri, 1941*). In addition, yield losses of up to 50% Mediterranean, Black Sea and Southwest Anatolia regions were reported (*Oran &*

*Parlak, 1969*; Özbaş, *1967*). Several studies were conducted to determine the prevalence of the disease from 1993 to 1997 and the results showed that its frequency changed from 4.9% to 28% (*Düşünceli, Çetin & Albustan, 2000*; *Mamluk et al., 1997*; *Yıldırım et al., 2000*). In addition, 21 races of the pathogen were reported in Türkiye (*Mert et al., 2012*) and the stem rust race, TTTTF, was commonly detected in Kastamonu province (*Akci & Karakaya, 2021*). As indicated above, stem rust disease have frequently occurred in some regions of Türkiye. However, the gene content of bread wheat varieties (whether they carried which *Sr* gene(s)) has not been identified. The objectives of this research were (i) to evaluate 150 bread wheat varieties for resistance to stem rust in adult plants under natural infection in the 2019–2020 and 2020–2021 growing seasons and (ii) to identify the presence or absence of resistance genes *Sr22, Sr24, Sr25, Sr26, Sr31, Sr38, Sr50,* and *Sr57* using molecular markers.

## MATERIALS AND METHODS

### Plant materials

In this study, 150 bread wheat (*Triticum aestivum* L.) varieties were analyzed. Information on these wheat varieties is provided in Table S1. In addition, the wheat cultivar "Morocco" was used as the susceptible control in the field trials in 2019–2020 and 2020–2021 growing seasons. Furthermore, the positive control lines, Prelude*6//Pre/Mq8*/Agent for *Sr24*, Eagle with *Sr26* for *Sr26*, Benno *Sr31*/6*LMPGfor *Sr31*, VPM (PI 519303) for *Sr38*, Fed*3/Gabo*51 for *Sr50*, Opata 85 for *Sr57*, were used.

### Assessment of field response to stem rust

Field tests were carried out to evaluate adult plant resistance of the varieties in the natural infection conditions at the experimental station of the Akdeniz University, Antalya, Türkiye (36°53′56″N 30°38′13″E). In November of both years, two rows measuring 100 cm in length were planted for each variety at the experimental site. Furthermore, the susceptible wheat variety "Morocco" was also sown as a spreader between the rows to assess rust reactions under natural infection conditions. Standard cultural practices were carried out at the site as well.

To evaluate the disease reactions at the adult plant stage (Zadoks scale 70 to 80), when the disease development reached 70% or more in the susceptible control, the disease was scored two to four times in field experiments at 7–10 days intervals. In field observations, the highest infection rate and disease severity score for each wheat variety were considered. The modified Cobb scale was used to determine the disease severity by estimating the proportion of the stem area covered by rust pustules (*Peterson, Campbell & Hannah, 1948*). Infection reaction was scored according to the size of pustules and quantity of necrosis and chlorosis on the stem described by *Roelfs (1992)*. The reactions were classified as: "0" denotes no visible symptoms, "R" denotes resistant expressed as no uredia and distinct necrotic area, "MR" denotes moderately resistant expressed as small uredia surrounded by either chlorotic and necrotic areas, "MS" denotes moderately susceptible expressed as medium uredia and surrounded by chlorotic areas and "S" denotes fully susceptibility expressed as uredia largely and no chlorotic and necrotic areas.

The coefficient of infection (CI) was calculated using the combined value of disease severity and reactions on an assigned scale. This scale was pointed out as: immune = 0.0, R = 0.2, MR = 0.4, MS = 0.8 and S = 1.0 (*Stubbs et al., 1986*). CI values were grouped as described by *Akan et al. (2012)*.

### DNA extraction and molecular detection of Sr genes

Total genomic DNA was extracted from seedlings of each variety at two leaf stages using the NucleoSpin® Plant II Extraction Kit (Macherey-Nagel, Düren, France) following the procedure described in the manufacturer's instructions. The quality and quantity of the extracted DNA was analyzed on 1% agarose gel electrophoresis stained with ethidium bromide. Then, they were dissolved with Tris-EDTA (TE) buffer at a strand concentration of 50 ng/μl and kept at −20 °C until using the molecular studies. The genomic DNAs of each variety were analyzed to identify of resistance genes *Sr24*, *Sr25*, *Sr26*, *Sr31*, *Sr38*, *Sr50*, and *Sr57* using different molecular markers. Information related to these markers is given in Table 1.

Total PCR volume of each primer was prepared as 15 μl consisting of 1 × PCR buffer, 1.5 mM $MgCL_2$, 0.2 mM each of the dNTP mixture, 1 μM each of forward and reverse primer, 1 U Taq DNA polymerase, 100 ng of DNA and 8.28 μl of double distilled water in a final volume of 15 μl. Amplification of DNAs was performed in a thermal cycler (T100; Bio-Rad, Hercules, CA, USA) under the following conditions except for *Sr24*#12 primer pair: 94 °C for 3 min for initial denaturation; 35 cycles of 94 °C for 30 s, 50 to 65 °C 30 s at annealing temperature depending on molecular markers (Table 2) for 1 min, and 72°C for 1 and 5 min of final extension at 72 °C. *Sr24*#12 was amplified in a touch-down PCR program by the following conditions: 7 cycles, 0.5 °C down each cycle starting from 62 °C and the remaining 30 cycles at 59 °C. Besides, each PCR analysis was performed two times to obtain accurate results.

The amplified PCR products were separated by 2% agarose gel electrophoresis at 80V for 1 h. The gels stained with ethidium bromide were visualized under UV light using a-gel imaging system (UVsolo Touch, Analytik, Jena, Germany). In addition, a 100 bp DNA Ladder (Thermo Fisher Scientific, Waltham, MA, USA) was used as a molecular weight marker.

### Statistical analysis

All obtained data were first recorded in Microsoft Excel. Basic statistical parameters (mean, minimum, maximum, coefficient of variation, standard deviation, skewness, and kurtosis) and correlation analysis were performed using Minitab® 21.4.1 software.

## RESULTS

### Evaluations of bread wheat varieties for stem rust resistance at adult plant stage

Among the varieties, disease severity rates ranged from 0.00 to 100, 0.00 to 90.00 in the growing seasons 2019–2020 and 2020–2021 respectively (Table 2). The overall coefficient of variation was calculated to be 123.55%. In addition, kurtosis and skewness values also

**Table 1 Information about molecular markers linked with stem rust resistance genes used in the present study.**

| Gene | Marker | Primer sequence (5′→3′) | Tm (°C) | Base pair (bp) | Reference |
|------|--------|--------------------------|---------|----------------|-----------|
| Sr22 | WMC633-F | ACACCAGCGGGGATATTTGTTAC | 56 | 117 | *Olson et al. (2010)* |
|      | WMC633-R | GTGCACAAGACATGAGGTGGATT | | | |
| Sr24 | Sr24#12 F | CTCACGCATTTGACCATATACAACT | 65 | 500 | *Mago et al. (2005)* |
|      | Sr24#12 R | TATTGCATAACATGGCCTCCAGT | | | |
| Sr25 | GBF | CATCCTTGGGGACCTC | 50 | 130 | *Prins et al. (2001)* |
|      | GBR | CCAGCTCGCATACATCCA | | | |
| Sr26 | Sr26#43-F | AATCGTCCACATTGGCTTCT | 56 | 270 | *Mago et al. (2005)* |
|      | Sr26#43-R | CGCAACAAAATCATGCACTA | | | |
| Sr31 | Iag95-F | CTCTGTGGATAGTTACTTGATCGA | 55 | 1,100 | *Mago et al. (2002)* |
|      | Iag95-R | CCTAGAACATGCATGGCTGTTACA | | | |
| Sr38 | VENTRIUP | AGGGGCTACTGACCAAGGCT | 65 | 262 | *Helguera et al. (2003)* |
|      | LN2 | TGCAGCTACAGCAGTATGTACACAAAA | | | |
| Sr50 | Sr50-5p-F3 | TTCAGTGAAGTTGCCGCTGT | 55 | 470 | *Mago et al. (2015)* |
|      | Sr50-5p-R2 | GCATGCTCTCAAGCTCCTTCT | | | |
| Sr57 | L34DINT9F | TTGATGAAACCAGTTTTTTTTCTA | 51 | 517 | *Lagudah et al. (2009)* |
|      | L34PLUSR | GCCATTTAACATAATCATGATGGA | | | |

confirmed a normal distribution for disease severity among the varieties (Table 2). Statistical analysis showed high correlations between the reactions of all varieties during the two growing seasons (Fig. 1). Based on the coefficient of infection (CI), the tested 150 bread wheat varieties were clustered into five groups: immune, expressed CI value to 0; resistant, expressed CI value 1 to 5; moderately resistant, expressed CI value 6 to 20; moderately susceptible, expressed CI value 21 to 60; and susceptible, expressed CI value 61 to 100. Among them, 94 (62% of varieties) were resistant or moderately resistant, whereas 44 (29%) showed moderately susceptible or susceptible reactions at the adult plant stage in 2020. In contrast, 85 (56% of varieties) were resistant or moderately resistant, but 53 (35%) were moderately susceptible or susceptible in 2021. Moreover, 12 varieties showed immune reactions in both years (Table S2).

## Molecular identification of *Sr* genes

In this study, six out of the eight *Sr* genes, *Sr22, Sr25, Sr31, Sr38, Sr50* and *Sr57* genotyped with molecular markers were identified in the 150 bread wheat varieties, whereas the resistance genes *Sr24* and *Sr26* were not detected in these varieties (Table S2). The SSR marker WMC633, tightly linked to *Sr22* at distances of 1–2 cm was used to determine the present gene (*Olson et al., 2010*). The marker produced several fragments. Among them, a fragment of 117 bp was evaluated as resistance, whereas other fragments of 171, 193, and 221 bp were susceptible. In the molecular screening of the tested varieties, only one variety "Kırgız 95" produced an expected fragment of 117 bp (Fig. S1). STS marker *Sr24#12*, which is completely linked to *Sr24*, was used to detect the presence of this gene (*Mago et al.,*

**Table 2 Basic statistical parameters of reactions of all varieties in 2 years and overall.**

| Season | N | Mean | Minimum | Maximum | CV (%)* | SD** | Kurtosis | Skewness |
|---|---|---|---|---|---|---|---|---|
| 2019–2020 | 150 | 19.63 | 0.00 | 100.00 | 140.94 | 27.67 | 0.82 | 1.47 |
| 2020–2021 | 150 | 22.21 | 0.00 | 90.00 | 112.25 | 24.93 | −0.25 | 1.09 |
| Overall | 150 | 20.92 | 0.00 | 95.00 | 123.55 | 25.85 | 0.27 | 1.28 |

**Note:**
* CV, coefficient of variation.
** SD, standard deviation.

*2005*). This marker is dominant and produces only a 500 bp fragment. According to the PCR results, *Sr24* was not detected among the tested varieties.

Molecular marker GB was used to determine the presence of *Sr25* (*Prins et al., 2001*). Six varieties of bread wheat (Nacibey, Ekiz, Murat-1, Bezosta-ja-1, Beşköprü, and Bone de) were found to carry the *Sr25* gene (Fig. S2). Another resistance gene, *Sr26*, is linked to the *Sr26#43* marker (*Mago et al., 2005*). Specific fragments of 207 bp related to the marker were not amplified among all the tested varieties. The resistance gene *Sr31* was identified using the molecular marker Iag95, which produces a 1,100 bp specific PCR fragment (*Mago et al., 2002*). Based on the molecular results, only one variety (Karacabey 97) carried *Sr31* (Fig. S3).

To identify the presence of *Sr38*, the marker VENTRIUP-LN2 was used to produce a 262 bp PCR fragment (*Helguera et al., 2003*). Molecular findings showed that only one variety (Çetinel 2000) carried this resistance gene (Fig. S4). Nine wheat varieties out of all tested wheat materials (Harmankaya-99, Lancer, Doğu 88, Palandöken 97, Meta 2002, Yüreğir-89, Özkan, Alka and Iridium) were detected to carry in the presence of the wheat stem rust resistance gene*Sr50* which produced a 470 bp band in PCR using the *Sr50*-5p F3/ *Sr50*-5p R3 primer pairs (Fig. S5). The primer pair, L34DINT9F/L34PLUSR, was used to determine the presence of the resistance gene *Sr57* (*Lagudah et al., 2009*) and 517 bp fragments were obtained by PCR amplification. According to the molecular results, 103 of 150 bread wheat varieties carried this gene (Fig. S6).

In this study, nine bread wheat varieties carrying combinations of two *Sr* genes were identified (Table 3). In this context, four wheat varieties (Doğu 88, Meta 2002, Özkan and Iridium) carried both *Sr50* and *Sr57*; three varieties (Ekiz, Bezostaja-1 and Beşköprü) contained *Sr25* and *Sr57*; one variety (Çetinel 2000) carried both *Sr38* and *Sr57*, and one variety (Karacabey-97) carried both *Sr31* and *Sr57*.

## DISCUSSION

Stem rust is a serious problem and yield loss because of its recently increased threat to wheat production. The main reason for these losses is the emergence of new virulent races, and the urediniospores of the disease can be dispersed over long distances by human activities or wind. Therefore, monitoring the changes in virulence of *Pgt* races is crucial. In addition, there is a need for a better understanding of the genetic resistance of wheat varieties. The resistance of varieties to stem rust under natural infection generally varies based on weather conditions and rust populations. The frequencies of the resistance groups R and MR were 56–62% in 2019–2020 and 2021–2021 growing seasons,

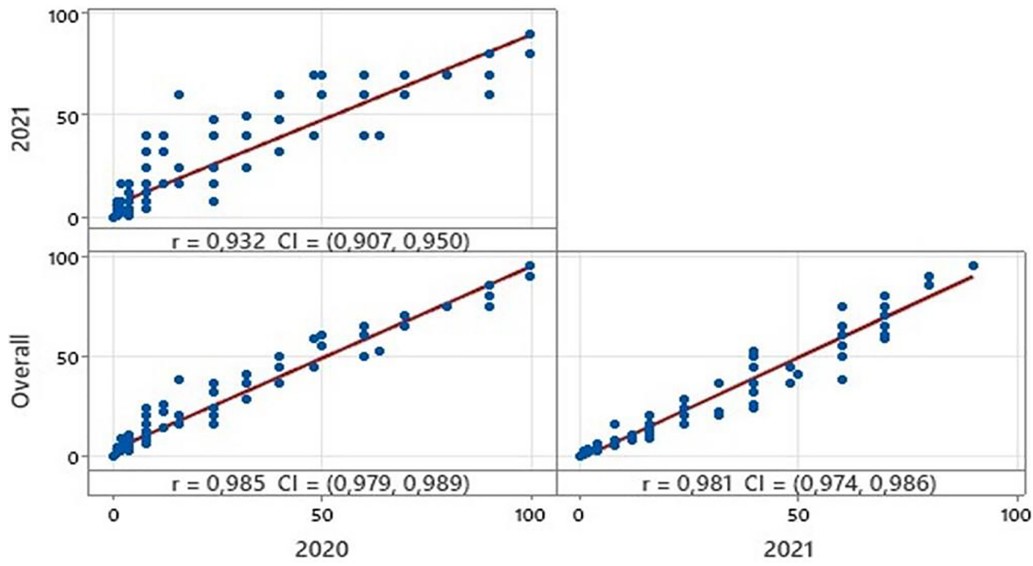

**Figure 1** Correlations between disease reactions of all bread wheat varieties in two growing seasons and overall.               

respectively. *Shamanin et al. (2016)* indicated that adult plant resistance to local stem rust population was observed in 26 genotypes (16.5%) of tested 146 wheat genotypes.
In another research conducted by *Elkot et al. (2020)* 12 of 38 Egyptian wheat varieties showed immune reactions under natural infection conditions during 2018–2019 growing period.

In the current study, the resistance of bread wheat varieties to stem rust disease under natural infection conditions was evaluated, and the major *Sr* genes were identified in these varieties. *Sr22* gene is derived from *Triticum monococcum* ssp. (*Gerechter-Amitai et al., 1971*) and *T. monococcum* L. (*Kerber & Dyck, 1973*) and is located on 7AL. In this study, the closest marker WMC633 flanking *Sr22* (*Olson et al., 2010*) was used to detect and it was found that "Kırgız 95" variety carried this gene.

Stem rust resistance genes *Sr24*, *Sr25*, and *Sr26* derived from *Thinopyrum ponticum* are commonly used in wheat breeding. *Sr25*, and *Sr26* provide resistance to most races of stem rust including Ug99 and its variants. The molecular markers *Sr24#12* linked with *Sr24*, Gb linked with *Sr25*, and *Sr26#43* linked with *Sr26* were used to identify these genes. None of varieties tested in this study carried either *Sr24* or *Sr26*. These findings are consistent with the study conducted by *Xu et al. (2018)*, who reported that 136 wheat varieties collected from China were molecularly screened to detect *Sr24* and *Sr26*, and their results showed that none of these varieties had two resistance genes. Similarly, *Sr26* was not detected in 119 wheat samples (*Li et al., 2016*). Moreover, similar findings shown by many researchers indicated not detected *Sr26* gene in the wheat genotypes in China (*Lin et al., 2021*; *Xu et al., 2018*, *2017*). As previous studies have reported, the combination of the *Sr25* gene with the *Sr6Agi* gene is widely preferred in wheat breeding to stem rust. However, *Baranova et al. (2023)* indicated that wheat varieties carrying only *Sr25* were susceptible to *Pgt*. In a study conducted by *Xu et al. (2018)*, two wheat varieties carried this gene pedigree information.

**Table 3 Number of bread wheat varieties including either each Sr gene or gene combinations by molecular results.**

| No. of varieties | Resistance genes | | | | | | | |
|---|---|---|---|---|---|---|---|---|
| | *Sr22* | *Sr24* | *Sr25* | *Sr26* | *Sr31* | *Sr38* | *Sr50* | *Sr57* |
| 1 | – | – | – | – | – | + | – | + |
| 4 | – | – | – | – | – | – | + | + |
| 3 | – | – | + | – | – | – | – | + |
| 1 | – | – | – | – | + | – | – | + |
| 141 | – | – | – | – | – | – | – | – |
| Total | 1 | 0 | 6 | 0 | 1 | 1 | 9 | 103 |

**Note:**
The presence (+) or absence (−) of *Sr* genes according to the molecular results.

This finding is consistent with another study by *Lin et al. (2021)*, which revealed that no wheat lines containing *Sr25* were detected among the varieties tested in their study. In contrast, this gene was found in six Turkish bread wheat varieties in this study.

The resistance genes *Sr31* and *Sr50* were transferred from the rye varieties "Petkus" and "Imperial", respectively. These two genes were introgressed into the wheat genome by translocation from1R (*Mago et al., 2015*). In this study, the *Sr31*-linked marker Iag95 and *Sr50*-linked marker *Sr50*-5p-F3, R2 were used to screen the bread wheat varieties in the present study. *Pretorius et al. (2000)* and stated that *Sr31* gene was not effective to Ug99. In 2014, the *Sr50* virulence isolate was found in a greenhouse, but this gene was still effective against Ug99 and its derivatives. According to the molecular results, one variety carried *Sr31* and nine varieties carried *Sr50*. On the other hand, the resistance gene *Sr31* was previously identified in the varieties Seri 82, Yildiz 98, Tahirova 2000, and Osmaniyem by *Yediay et al. (2010)*. According to the results of the present study, Seri 82, Yildiz 98, Tahirova 2000, and Osmaniyem do not carry *Sr31*. This result can be attributed to the utilization of different DNA primers in these studies.

The resistance gene *Sr38*, which is linked with other rust resistance genes, namely *Yr17* and *Lr37* derived from *T. ventricosum*, is widely preferred because of its resistance to multiple diseases. The gene *Sr57* associated with resistance to wheat yellow rust (*Yr18*), leaf rust (*Lr34*), and powdery mildew (*Pm38*) provides broad-spectrum resistance to these wheat pathogens and is effective in adult plants. *Olivera et al. (2022)* detected *Sr* genes in 120 bread wheat varieties and breeding lines in Spain *via* molecular markers. They reported that *Sr38* (in 50 varieties), *Sr31* (in 16 varieties), *Sr24* (in one variety), *Sr7a* (in 11 varieties), and *Sr57* (in 14 varieties). *Lin et al. (2021)* indicated that the *Sr38* gene was detected in 19 of 95 tested wheat lines. These results are correlated with the findings reported by *Xu et al. (2018)*, who detected the *Sr38* gene in 28 of 136 wheat varieties in China. *Kelbin et al. (2022)* demonstrated that *Sr57* is present in 17 of 92 new Siberian agricultural wheat germplasms. In contrast, the majority of the wheat varieties tested in this study carried the gene *Sr57*.

Overall, this study presents potentially useful information regarding the resistance of wheat varieties to stem rust disease in Türkiye. The effect of *Sr57* on CI deserves more

attention. The varieties with *Sr57* had 65% less stem rust than those without it. However, there is wide variation within the *Sr57* and non-*Sr57* groups, indicating that *Sr57* alone is not sufficient to achieve an acceptably high resistance. The implication for breeding is that selecting *Sr57* can help achieve moderately resistant to stem rust, but further selection in trials at well-infected sites is expected to further improve resistance, as shown by the many varieties with *Sr57* and low CI values. *Sr26* was not found in bread wheat varieties, despite its long-standing value as a source of durable resistance to stem rust. *Sr25*, *Sr38*, and *Sr50* did not control stem rust in the field, as indicated by the variety scores in the results of the current study. Only one variety each had *Sr22* and *Sr31*, so little can be said about the value of these genes. Both varieties had fairly low field scores but were not immune to stem rust. This finding can be explained by the fact that *Sr22* and *Sr31* are partially effective against avirulent pathogen population, or that virulence is at a moderate frequency in the pathogen population.

## CONCLUSIONS

In this study 150 wheat varieties commonly grown in different regions of the Türkiye were evaluated under natural infection conditions to *Pgt* disease and they were screened at molecular level using the gene-specific DNA markers to determine the stem rust resistance genes. The results showed that except for *Sr24* and *Sr26*, the remaining resistance genes were determined in the wheat varieties in different rates. The varieties carrying the same resistance genes have shown different reactions to stem rust, and this variability can be attributed to the presence of unidentified resistance genes, varying expression levels of them, and the influence of other stress factors. The knowledge obtained from the present study is a valuable tool for wheat breeders and wheat varieties carrying *Sr* gene or their combinations can be utilized in future breeding research aimed at enhancing stem rust resistance.

## ACKNOWLEDGEMENTS

The author thanks Dr. Mehmet Tekin from Akdeniz University in Türkiye for providing plant materials.

### Funding

The author received no funding for this work.

### Competing Interests

The author declares that they have no competing interests.

### Author Contributions

- Ahmet Cat conceived and designed the experiments, performed the experiments, analyzed the data, prepared figures and/or tables, authored or reviewed drafts of the article, and approved the final draft.

## Data Availability

The raw data are available in the Supplemental Files.

## Supplemental Information

Supplemental information for this article can be found online at http://dx.doi.org/10.7717/peerj.17633#supplemental-information.

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
