# Peer review of "Evaluation of genetic variation and host resistance to wheat stem rust pathogen (Puccinia graminis f. sp. tritici) in bread wheat (Triticum aestivum L.) varieties grown in Türkiye"

_PeerJ, doi:10.7717/peerj.17633_

## Round 0.1 · original submission · Major Revisions

Dear authors,

According to the reviewer's comments, your article requires major revisions. Please refer to the reviewer's comments and make revisions accordingly.

**Language Note:** The review process has identified that the English language must be improved. PeerJ can provide language editing services - please contact us at [email protected] for pricing (be sure to provide your manuscript number and title). Alternatively, you should make your own arrangements to improve the language quality and provide details in your response letter. – PeerJ Staff

Reviewer 1 ·

Basic reporting

Dear editor,
Thank you for the opportunity to review this Manuscript - Genetic variation in bread wheat (Triticum aestivum L.) varieties for resistance to wheat stem rust pathogen (Puccinia graminis f. sp. tritici). Wheat stem rust, caused by the fungus Puccinia graminis f. sp. tritici (Pgt), poses a substantial threat to global wheat production. In this study, the resistance of 150 bread wheat varieties was evaluated by the natural Pgt infection at the adult-plant stage during 2021–2022, and molecular testing were used to identify the presence of 8 Sr genes in these cultivars. These study findings will significantly contribute in future wheat pre-breeding. The manuscript still requires a lot of careful revision before publication.

Experimental design

no comment

Validity of the findings

This study is important for breeding wheat cultivars with resistance to stem rust

Additional comments

The manuscript still requires a lot of careful revision before publication.
The main issues are:
1.The title of the paper does not present the content of the article well. It is recommended to revise it.
2.The author used natural Pgt infection method to inoculate wheat stem rust, so we do not know the virulence spectrum of natural Pgt and the amount of natural inoculation during 2021–2022. The best way is to add some wheat lines with known Sr genes during field testing. It is also possible that the author does not have such wheat lines.
3.The discussion part needs to be rewritten. In particular, only one of the 150 wheat varieties contains Sr31. As we all know, this gene is the most widely used in wheat breeding in the world, and why there are so few wheat varieties in Türkiye should be analyzed and discussed. In addition, the remaining genes also need to be discussed and analyzed based on the variety pedigree.
4.The effectiveness of molecular markers may vary depending on the background of different wheat varieties. Therefore, negative controls for molecular testing should include the highly susceptible wheat variety Morocco. You can refer to two references: Wu et al. Journal of Integrative Agriculture, 2023, 22(6): 1740–1749;Sun et al, Front. Plant Sci., 2023, 14:1156936
Others
L41: All Sr genes must be italicized and modified throughout the text.
L59-60: Add references to “It was reported that yield losses of up to 100% have occurred during epidemic conditions”.
L60-63:The United States and Europe are the main countries that use the eradication of Berberis 61 spp. as an effective method to control wheat stem rust, rather than most other countries.
L65: renamed as TTKSK
L66: The cited literature comes from 2008, and so far, there must be over 17 genes that are ineffective against Ug99.
L68: changed to ...Sr24, Sr36, Sr38, and SrTmp
L72:In 2016, the stem rust race TTTTF caused the breakdown of resistance in numerous durum and bread wheat varieties in Sicily, representing a significant threat that has been identified in other European countries. You should add information about this new race.
L75-76: To avoid this, the gene content of wheat 76 varieties must be knowledge and their prevalence. Rewrite.
L80: more than 70 stem rust resistance genes
L81-83: Among those used in this study, seven genes namely Sr22, Sr24, Sr25, Sr26, Sr31, Sr38 and Sr57 confer all-stage resistance to Ug99 and the gene Sr57 is effective in the adult plant stage (APR). Rewrite. From this, it can be seen that you are a beginner in the study of wheat stem rust disease. Ug99 has attracted high attention due to its virulence to Sr31.
L91: from 40% to 75%
L95: from 4.9% to 28%
L96: Whether wheat stem rust has occurred in Türkiye in recent years, because the latest reference you cited is 2000. If yes, please add relevant information.
L105: Are these 150 wheat varieties the main varieties in production in Türkiye?
L120: At the beginning of recording data, what is the growth period of wheat?
L125: described
L119: I suggest changing the disease reactions or reactions throughout the entire text to infection types
L146: changed “µl” to “µL”

Reviewer 2 ·

Basic reporting

Unfortunately, the English language of the manuscript needs improving. I recommended that the paper to be checked by a fluent English speaker or professional English editing service.
Some sentences are even difficult to understand.
For example, in lanes 46 and 47 (abstract):
‘However, Sr24 and Sr26 were determined none of them. These resistance genes were identified for the first time during this study.’ What does the latter sentence mean?
The introduction, in general substantiates the importance of the current study, but some revisions are to be made.
In lanes 81-83, Sr57 is referred to as both all-stage resistance genes and APR; moreover, Sr31 is listed among genes conferring resistance to Ug99. ‘Among those used in this study, seven genes namely Sr22, Sr24, Sr25, Sr26, Sr31, Sr38 and Sr57 confer all-stage resistance to Ug99 and the gene Sr57 is effective in the adult plant stage (APR).’
Sr57 is APR, and Sr31 does not confer resistance to Ug99!!! In addition, there are Ug99 biotypes virulent to Sr24 (Jin Y, Szabo LJ, Pretorius ZA, Singh RP, Ward R, Fetch T Jr. Detection of Virulence to Resistance Gene Sr24 Within Race TTKS of Puccinia graminis f. sp. tritici. Plant Dis. 2008 Jun;92(6):923-926. doi: 10.1094/PDIS-92-6-0923.) This part of introduction should be revised.
In general, the structure of the article confirms to PeerJ standards, the figures are of sufficient quality, the raw data are available. Please write gene names (Sr31 etc.) in italic.

Experimental design

Stem rust is a dangerous disease affecting wheat yield, so the evaluation of stem rust resistance of varieties is an important problem. Molecular detection of Sr genes provides information to increase the efficiency of breeding using MAS. As the natural infection background of stem rust is high in Türkiye, some Turkish bread wheat varieties may be an important source of resistance genes for breeders worldwide. Because of this the subject of this manuscript is of importance.
Some details are to be provided or revised in Materials and Methods.
Lane 109. Please add the information about the positive controls for Sr22 and Sr50.
In the section Assessment of field response to stem rust (lane 111), please add GPS coordinates of the experimental plot and the dates of sowing in both years.
Lane 143 as well as discussion, lane 265: Sr57 is not linked to ‘resistant genes for another rust pathogen’. The same gene (Lr34/Yr18/Pm38/Sr57/Bdv1) confers moderate resistance to stem, leaf and yellow rusts, as well as powdery mildew and barley yellow dwarf virus. So these parts of the text are to be revised.
In Table 1, the correct reference for primers to Sr57 is Lagudah et al., 2009 (Lagudah ES, Krattinger SG, Herrera-Foessel S, Singh RP, Huerta-Espino J, Spielmeyer W, Brown-Guedira G, Selter LL, Keller B. Gene-specific markers for the wheat gene Lr34/Yr18/Pm38 which confers resistance to multiple fungal pathogens. Theor Appl Genet. 2009 Sep;119(5):889-98. doi: 10.1007/s00122-009-1097-z).
Lanes 151, 152,’ Amplification of DNAs was performed in a thermal cycler (T100, Bio-Rad, USA) under the following conditions except for Sr24#12 primer pair: 94°C for 3 min for initial denaturation; 35 cycles of 94°C for 30 s, 50 to 65°C 30 s at annealing temperature depending on molecular markers (Table 2) for 1 min,..’. What was the correct time at the annealing stage, 30 s or 1 min?
Please present the data for the variety Morocco in Table S2.

Validity of the findings

There are some questions to molecular identification of certain Sr genes.
Bezostaja-1 (Bezostaya -1) is a much studied variety developed in the former USSR, and it seems hardly probable that it carries Sr25/Lr19 (for example, Lr19 was not detected in Bezosaya-1 by Kokhmetova, A., Madenova, A., Kampitova, G. et al. Identification of Leaf Rust Resistance Genes in Wheat Cultivars Produced in Kazakhstan. CEREAL RESEARCH COMMUNICATIONS 44, 240–250 (2016). https://doi.org/10.1556/0806.43.2015.056).
The most problematic issue is identification of varieties carrying Sr50. In FigS5, the size of the marker band, which is to be 470 bp, is larger than 500 bp, and there is no positive control on the gel. Moreover, the varieties Doğu 88, Palandöken 97, Harmankaya-99, and Yüreğir-89 were among Turkish varieties and lines tested with nine molecular markers for the presence of wheat-rye translocations (markers for arm 1RS) by Yediay et al. (2010), and no rye material was found in these four varieties (Yediay, F.E., Baloch, F.S., Kilian, B. et al. Testing of rye-specific markers located on 1RS chromosome and distribution of 1AL.RS and 1BL.RS translocations in Turkish wheat (Triticum aestivum L., T. durum Desf.) varieties and landraces. Genet Resour Crop Evol 57, 119–129 (2010). https://doi.org/10.1007/s10722-009-9456-9). Contrary to the wheat-rye translocations from the rye Petkus (1BL.1RS) and Insave (1AL.1RS), the 1DL.1RS translocation from the rye Imperial was not reported to be present among commercial wheat varieties. It is hardly probable that 1DL.1RS with Sr50 is present in Turkish varieties developed more than 20 years ago. So the identification of varieties with Sr50 seems ambiguous and if the proper gels with 470-bp amplicons are lacking, this part of molecular analysis could be omitted from the present manuscript.
Another ambiguous item is identification of varieties with Sr31 (on the 1BL.1RS translocation from Petkus). 1BL.1RS (with Sr31) was previously identified in the varieties Seri 82, Yildiz 98, Tahirova 2000, and Osmaniyem by Yediay et al. (2010). According to the results of the present study, Seri 82, Yildiz 98, Tahirova 2000, and Osmaniyem do not carry Sr31 (Table S2).
As a tip, the presence of any translocation involving arm 1RS can be easily and quickly checked by acid polyacrylamide gel electrophoresis of seed proteins from the presence of a group of secalins.
So please revise the text considering the above items (in both Results and Discussion) and include the reference Yediay et al. (2010).
Lane 258. ‘The Sr31 is an effective Ug99 race of the stem rust.’ This sentence is problematic in both grammar and sense. Sr31 is not effective to Ug99! Please revise correspondingly the paragraph (lanes 256-262).
An important part of this investigation is analysis of the presence of Sr57. These results deserve more attention. I suggest to describe the frequencies of cultivars that showed stem rust resistance, moderate resistance and susceptibility within each group and compare them more precisely (in Results and in Discussion). This is mentioned only in lane 277, but the sentence should be revised.
The section Conclusions is to include conclusions but not a summary (sentences on lanes 291-295 should be removed). Please revise the conclusions on lanes 296-300, as Sr31 was previously identified by Yediay et al. (2010) (1BL.1RS) and also mention the high frequency of Sr57.

Reviewer 3 ·

Basic reporting

Since wheat stem rust, especially its new races, is a more destructive disease of wheat on a global scale, the reviewed manuscript is characterized by a sufficient degree of relevance. The author analyzes the presence of stem rust resistance genes in a large sample of wheat varieties grown in Turkey (150 varieties). The work was carried out at a fairly good experimental level using high-quality molecular genetic markers. As result of conducted experiments, some selected wheat varieties with useful resistance to stem rust can be used for breeding of new genotypes with better durable durable resistance to stem rust.

Experimental design

However, a detailed reading of the text of the article does not allow us to get an answer why the author chose this particular set of genes for research: resistance genes (Sr22, Sr24, Sr25, Sr26, Sr31, Sr38, Sr50 and Sr57)? If we rely on one of the modern reviews on this issue (https://doi.org/10.3390/pathogens11101157), the author did not study own race-specific stem rust resistance genes in bread wheat and only one own race-nonspecific stem rust APR gene of common wheat (Sr57 from Bezostaya 1), which provides resistance to Ug99. Among other stem rust resistance genes, introgressed from the relatives (including wild) (Sr22, Sr24, Sr25, Sr26, Sr31, Sr38, Sr50), Sr31 and Sr38 do not confer respective resistance. This question must be considered by author and respective explanations/argumentation should be incorporated in the text.

Respectively, we ask to explain why author did not used direct primers for sequenced genes and did not tested the presence of such genes as Sr33 (DOI: 10.3103/S009545272306004X) and Sr35 (DOI: 10.1126/science.123902) important for Ug99 stem rust resistance,

Validity of the findings

No comments

Additional comments

No comments

---

## Round 0.2 · Major Revisions

Dear authors,

We have received the evaluations from our peer reviewers, and after careful consideration, we believe that your work holds potential but requires further revision to meet our publication standards. Particularly, we draw your attention to the comments provided by Reviewer 3. Their insights are crucial for enhancing the quality and impact of your manuscript. To proceed, we kindly ask you to meticulously address each point raised by Reviewer 3 and comments from the other two reviewers.

Please revise your manuscript according to the reviewers' suggestions and provide a point-by-point response to each comment. This should include how you have incorporated their feedback or a reason if any comment was not followed.

Best regards,

Yunfeng Xu

Reviewer 1 ·

Basic reporting

Dear editor,
The author made revisions to the manuscript according to my suggestions, but they still need to be carefully revised before publication.
1. The effectiveness of molecular markers may vary depending on the background of different wheat varieties. Therefore, negative controls for molecular testing should include the highly susceptible wheat variety Morocco.
2. All Sr genes must be italicized and modified throughout the text. Although the author stated that all modifications have been made, there are still many genes in the article that have not been italicized. eg. L40, Sr22; L153, Lr34/Yr18/Pm38/Ltn1; L197, Sr24;
3. L2: Sr genes;
4. There are multiple places in the text with two commas or periods;
5. L98: in wheat
6. L117: rewritten
7. L189: “Molecular identification of Sr genes” starts a new line
8. L231:56%-62%
9. L240: from Triticum monococcum ssp.
10. L244: Chang “Sstem" to “Stem”; Sr26 is not commonly used in wheat breeding
11. L267-269: “On the other hand, the resistance gene Sr31 was previously identified in the varieties Seri 82, Yildiz 98, Tahirova 2000, and Osmaniyem by Yediay et al. (2010). According to the results of the present study, Seri 82, Yildiz 98, Tahirova 2000, and Osmaniyem do not carry Sr31.” Why?
12. L232: delete “These results indicated that the pathogen population in both years was highly virulent.”
In addition, there should be a fluent English speaker to edit the manuscript before publication.

Experimental design

no comment

Validity of the findings

This study is important for breeding wheat cultivars with resistance to stem rust

Additional comments

no comment

Reviewer 2 ·

Basic reporting

The manuscript has been significantly improved and I agree with nearly all revisions.
The only item with which I’m not fully satisfied is the response to my previous comment. ‘In FigS5, the size of the marker band, which is to be 470 bp, is larger than 500 bp, and there is no positive control on the gel’.
Response: Some of the bands from the gel image appear to be on above, but the amplified product is about 470 bp.
I’m relying on the Editor’s decision in this case.
Nevertheless, some minor corrections are to be made.
Please revise the sentences in lanes 60-63: it seems that in lane 60 some phrases are omitted after ‘varieties’ and in lane 62 the beginning of the sentence was lost.
Please replace the reference in lane 272 (Baranova et al (2023)) by Pretorius et al. (2000), as it is Pretorius et al. who first detected that Sr31 was not effective to Ug99.
Pretorius, Z.A.; Singh, R.P.; Wagoire, W.W.; Payne. T.S. Detection of virulence to wheat stem rust resistance gene Sr31 in Puccinia graminis f. sp. tritici in Uganda. Plant Disease 2000, 84(2), 203.
Lane 281: please replace ‘linkage’ by ‘associated with resistance’
I would recommend to remove the phrase in lane 292 ‘When these valuable findings are evaluated,’ and start the sentence with ‘The effect...’
Please remove ‘P: Positive control,’ from the caption to Figure S2.
I believe the poofs will be thoroughly corrected, but I suggest some minor corrections
Lane 47: please remove ‘been’.
Lane 106: Please replace ‘was’ by ‘were’
Lane 122: please correct ‘2seasons019’
Lane 181: maybe ‘stage’ should be added (at adult plant stage)
Lane 193: It seems that ‘in 2021’ should be added after ‘susceptible’
Lane 261: Please revise the sentence.
Lane 273: Please replace ‘Sr31 gene was not resistance’ by ‘Sr31 gene was not effective’

Experimental design

no comment

Validity of the findings

no comment

Additional comments

no comment

Reviewer 3 ·

Basic reporting

I did not receive clear answers on my questions, especially question 2.
Moreover, it is very difficult to find changes/improvements in the text, because they are not underlined or highlighted.
So, I can not able to make positive decision.

Experimental design

I did not receive clear answers on my questions, especially question 2.
Moreover, it is very difficult to find changes/improvements in the text, because they are not underlined or highlighted.

Validity of the findings

-

Additional comments

I did not receive clear answers on my questions, especially question 2.
Moreover, it is very difficult to find changes/improvements in the text, because they are not underlined or highlighted.
So, I can not able to make positive decision.

---

## Round 0.3 · Minor Revisions

In addition to the remaining final issues from the reviewers, here are some additional considerations for the author:

Water is usually used as none-template control, rather than negative control. So, negative control such as susceptible variety Morocco is suggested to be included for genotyping.

Line 99: Several surveys studies

Line 352: two commas

All Sr, Lr gene names should be italicized. Check the manuscript thoroughly again, including the references.
Line 146: Sr
Line 218: Sr50
Line257: Sr6Agi

Line 230: Revise the sentence.

Reviewer 1 ·

Basic reporting

Dear editor,
The author made revisions to the manuscript according to editor or reviewers suggestions, but they still need to be carefully revised the language before publication.
1. L61-66, rewriting. “Some aggressive pathogen races renamed as TTKSK have recently been reported in the last two decades. For example, the resistance gene, Sr31, was broken down by the new emergent race Ug99 designated as TTKSK in Uganda in 1998 (Pretorius et al., 2000) and literature comes from 2008, and so far, there must be over 17 genes that are ineffective against Ug99 (Singh et al., 2008). It was later reported that new variants evolved from Ug99 and their virulence on resistance genes Sr24, Sr36, Sr38 and SrTmp (Jin et al., 2008; Jin et al., 2009; Newcomb et al., 2016).”
2. L71-73, delete “Wheat varieties carrying resistance genes are used to effectively control the environment of stem rust disease.”
3. L80, Add a “.” after (McIntosh et al., 2016)
4. L81-83, delete “In this study, Sr resistance genes, specifically Sr22, Sr24, Sr25, Sr26, Sr 31, Sr38, and Sr57 provide resistance to stem rust at all stages, with Sr57 demonstrating effectiveness during the adult plant stage (APR)”.
5. L88, change “the Pgt pathotype TTKSK” to “TTKSK”.
6. L120, Benno/6*LMPG-6 or Sr31/6*LMPG, please check.
7. L153, change to “…Sr38, Sr50, and Sr57…”
8. L154-155, delete “Additionally, Sr38 (Lr37/Yr17) and Sr57 (Lr34/Yr18/Pm38/Ltn1) are linked to resistant genes for another diseases.”
9. L233, change “Therefore, pathogen monitoring in wheat species and alternate hosts is crucial.” to “Therefore, monitoring the changes in virulence of Pgt races is crucial.”
10. L248-249, Sr24 is ineffective for variants of Ug99
11. L249, change to “The molecular markers Sr24#12…”
12. L251-256, The use of SR24 and Sr26 genes in wheat varieties in China is relatively limited, so why is it also limited in your country? You need to analyze whether these two genes are used in breeding in your country and the pedigree of the varieties. The results of other countries cannot be used as evidence for your research results.
13. The format of references needs to be unified
14. L131, What growth stage of wheat when the disease development reached 70%, please add
In addition, there should be a fluent English speaker to edit the manuscript before publication.

Experimental design

no comment

Validity of the findings

no comment

Additional comments

no comment

Reviewer 2 ·

Basic reporting

The author has made the major revisions.
Fig S5 is satisfactory.
However, some minor revisions are needed to improve the text.

Please revise the text in lanes 56-60, it seems that something was accidentally deleted.
‘In the past, growing resistant varieties and the United States and Europe are the main countries that use the eradication of Berberis spp. as an effective method to control wheat stem rust, rather than most other countries. (Peterson et al., 2005) except for Ethiopia where the pathogen caused dramatic yield losses of wheat variety “Enkoy” in 1993 and 1994 (Shank 1994; Singh et al., 2011).’

Lanes 62-64
‘For example, the resistance gene, Sr31, was broken down by the new emergent race Ug99 designated as TTKSK in Uganda in 1998 (Pretorius et al., 2000) and literature comes from 2008, and so far, there must be over 17 genes that are ineffective against Ug99 (Singh et al., 2008).

lane 69 add (TTRTF) after TTTTF:
‘2016, the stem rust race TTTTF (TTRTF)’

in lanes 76, 77, I’d recommend some changes:
resistance genes in wheat have been grouped into two groups, conferring forms all-stage or seedling resistance (ASR) and adult plant resistance (APR).

In lanes 81-82 I’d recommend some changes
varieties, varieties or landraces against stem rust disease (Goutam et al., 2015). So far, 70 stem rust resistance genes have been reported in wheat and their relatives (McIntosh et al., 2016). In this study, Sr resistance genes, anayzed in this study, specifically Sr22, Sr24, Sr25, Sr26, Sr 31, Sr38, and Sr57 provide resistance to stem rust at all stages, while with Sr57 demonstrates ing effectiveness during the adult plant stage (APR).

In lanes 105-106, I’d recommend some changes:
However, the gene content of bread wheat varieties (whether they carried which Sr gene(s)) has not been identified.

lane 155
replace ‘were’ by ‘is’

In lanes 198-200, I’d recommend some changes
Among them, a fragment of 117 bp was evaluated as a resistance marker resistant, whereas other fragments of 171, 193, and 221 bp indicated susceptibility were susceptible. Molecular screening of the tested varieties, only one variety “K1rg1z 95” produced an expected fragment on 117 bp (Figure S1).

In lane 217, I’d recommend some changes
Nine wheat varieties out of all tested wheat materials (Harmankaya-99, Lancer, Dogu 88, Palandöken 97, Meta 2002, Yüre ir-89, Özkan, Alka and Iridium) were detected to carry in the presence of the wheat stem rust resistance gene Sr50, as they which produced a 470 bp band in PCR using the Sr50-5p F3/Sr50-5p R3 primer pairs (Figure S5).

In lane 255, I’d recommend some changes
‘...shown by many researchers indicated the absence of the not detected Sr26 gene in the wheat genotypes in China (Lin...’

lane 160:
replace consisted by consistent

lane 267. please remove ‘Baranova et al.’ in this case

lane 311. Please revise the end of the sentence (‘....can be used in breeding studies that will be improved on stem rust resistance in the future’).

Experimental design

no comments

Validity of the findings

no comments

Additional comments

no comments

Reviewer 3 ·

Basic reporting

Finally, I satisfied with improvements and answers on my questions.

Experimental design

Good

Validity of the findings

Good

Additional comments

No

---

## Round 0.4 · accepted · Accept

Dear Dr. Cat,

Your manuscript is accepted for publication, but some very minor language editing is needed, as per my attached PDF, which can be performed while in production. These revisions are necessary to ensure the clarity and completeness of your manuscript.